# Effect of UFC on the Microscopic Pore Structure of Cemented Soil in Humic Acid Environment

**Jing Cao** [1], **Fangyi Liu** [1], **Siyang Huang** [1,*], **Hong Liu** [2], **Zhigang Song** [1], **Jianyun Li** [1] and **Guoshou Liu** [1]

[1] Faculty of Civil Engineering and Mechanics, Kunming University of Science and Technology, Kunming 650500, China
[2] Faculty of Art and Communication, Kunming University of Science and Technology, Kunming 650500, China
* Correspondence: huangsiyang@stu.kust.edu.cn

**Abstract:** Peat soil is widely distributed in the Dianchi Lake area of Yunnan, and the effect of the cement deep-mixing method on peat soil foundation is mainly affected by humic acid (HA). In this paper, a composite cement curing agent is formed by adding different proportions of ultra-fine cement (UFC) to ordinary Portland cement (OPC) and used to cure the HA-containing cohesive soil. Mercury intrusion porosimetry (MIP), scanning electron microscopy (SEM), and X-ray powder diffraction (XRD) are used to study the influence mechanism of UFC on the micropore structure of HA-containing cemented soil. The unconfined compressive strength test (UCS) is used to verify it. MIP, SEM, and XRD results show that UFC can significantly improve the microscopic pore structure of the samples. The hydration reaction rate of cement increases with the increase in the proportion of UFC, and the generated hydration products can fill the pores of the samples. The filling effect of hydration products on macropores is enhanced, and the pores change from fibrous filling to cemented filling. The enhanced cementation of the hydration products improved the loose and overhead structure inside the sample. Enhancing the cementation of hydration products improves the loose and overhead structure inside the sample and the integrity of cemented soil. UCS verified that the increase in the UFC proportion increases the HA-containing cemented soil strength. It achieves the purpose of reducing the amount of cement when curing peat soil foundations and supports reducing carbon emissions in practical projects.

**Keywords:** ultra-fine cement; cemented soil pore structure; humic acid; MIP; SEM; UCS

## 1. Introduction

Cement is one of the important building materials and belongs to powder materials. China is the largest cement producer in the world, and ordinary Portland cement is the most widely used in construction projects [1–3]. Cement as a curing agent is a necessary technical means of treating soft soil foundations. Because of its many advantages, the cemented soil mixed method is often used in engineering [4–7]. The Quaternary sediments in the Dianchi Lake area are deep and the widely distributed peat soil have brought many problems to the economic construction of Kunming [8–10]. Peat soil is a special soft soil composed of humic groups (HG), soil inorganic matter, and animal and plant residues. It has a large natural void ratio, high natural water content, high organic matter content, large compressibility coefficient, low shear strength, and low bearing capacity [11–14]. HG is mainly composed of humic acid (HA), fulvic acid, and humin and is a natural organic polymer mixture with a complex structure. The effect of HA on cemented soil is more potent than that of fulvic acid and humin [15,16]. Therefore, improving the peat soil foundation with insufficient bearing capacity by changing the cement properties has become an important research topic in the civil engineering industry.

The influencing factors of cement performance are mainly in two aspects [17–22]: (1) The chemical and mineral components affect the degree and speed of the hydration reaction of cement and (2) cement particles have powder characteristics, in which particle gradation and specific surface area significantly impact cement performance. Under the condition of not changing the cement composition, reducing the cement fineness is particularly important to improve cement performance. Ultra-fine cement (UFC) has smaller particles and a larger specific surface area than ordinary cement and has the characteristics of better fluidity under the same water consumption [23]. UFC is defined at home and abroad as follows: (1) UFC is made of the same material as ordinary Portland cement (OPC). (2) The maximum particle size of UFC is less than 20 $\mu m$, $d_{95} < 20$ $\mu m$ and the median particle size is less than 5 $\mu m$. (3) The specific surface area is greater than 8000 $cm^2/g$ [24–27]. Due to the high cost of UFC, it is not in line with engineering practice to completely use UFC in foundation treatment. Therefore, adding a certain proportion of UFC to cement and exploring its effect on the microscopic pore structure of HA-containing cemented soil has high theoretical value and practical significance.

Tsivilis et al. [28] proposed that particles of 3~32 $\mu m$ in cement play a major role in the strength of cement, and their mass proportion should account for more than 65%. Kontoleontos et al. [29] studied the effect of colloidal nano-silica on the hydration reaction of UFC and found that it can promote the pozzolanic effect and improve the microstructure of hardened cement paste through experiments. Wang et al. [30] found that in the case of the same raw materials, different processes can obtain cement with different levels of fineness, and its particle characteristics are also different, affecting cement's performance. Bentz [19] proposed that although the cement performance of OPC is lower than that of UFC, the hydration reaction of OPC releases less heat, and the risk of early cracking of concrete is lower. Zheng et al. [31] studied the effect of UFC on the early compressive strength of solidified soft soil. The results showed that with the decrease in cement fineness, the unconfined compressive strength and elastic modulus of solidified soft soil gradually increased. Huang et al. [32] studied the microstructure characteristics of cemented soil qualitatively and analyzed that improving the micropore structure of cemented soil would increase strength. Huang et al. [33] proposed that when the cement particle distribution is optimal, the hardened cement paste's porosity and pore size become smaller, increasing its strength. In summary, there are few related studies at home and abroad. It is of great significance to use the cemented soil mixing method to treat peat soil foundation by studying the effect of different proportions of UFC on the micropore structure of HA-containing cemented soil.

Pores in cemented soil occupy a certain space, and the size, quantity, connectivity, and distribution characteristics of pores are intrinsically related to the macroscopic properties of the consolidated body. Therefore, the strength properties of cemented soil can verify the effect of UFC on the pore structure of HA-containing cemented soil. The widely used methods in studying the micro-porosity of cemented soil are MIP, scanning electron microscopy (SEM), and X-ray powder diffraction (XRD). Microscopic tests study the effect of UFC on the microscopic pore structure of HA-containing cemented soil, and an unconfined compressive strength test (UCS) verifies the effect of UFC on HA-containing cemented soil. Combined with the results of the microscopic and strength tests, it is verified that adding UFC can increase peat soil's strength, thereby reducing the cement used in curing peat soil. Studies by many scholars [34,35] have proved that the direct carbon emissions in the process of cement production account for about one-quarter of the global industrial carbon emissions and about 0.8 t of $CO_2$ is emitted per 1 t of cement produced. Therefore, the cement mixing method is one of the technical means to strengthen the peat soil foundation, and this experimental study can provide certain theoretical support for reducing the amount of cement in practical projects. It has important practical significance for achieving China's carbon peaking and carbon neutrality goals.

This paper uses MIP, SEM, XRD, and UCS tests on the HA-containing cemented soil with different proportions of UFC. The effect of UFC on the microscopic pore structure of cemented soil in a humic acid environment is explored through the test results.

## 2. Materials and Methods

### 2.1. Experimental Materials

The soil used for the experiment is taken from the north slope of the Jingyuan, Chenggong Campus dormitory area, Kunming University of Science and Technology. The physical property indicators measured by the test are shown in Table 1. The undisturbed soil is yellow–brown cohesive soil with low organic matter content. The experiment used humic acid (HA) reagent produced by Tianjin Guangfu Chemical Reagent Factory. Relevant research results show that the carbon content of HA is about 60% [36,37], and combined with the actual test results of the carbon content of the HA reagent, the HA content is calculated to be 41.68%. Ordinary Portland cement (OPC) P·O42.5 produced by Huaxin Cement (Honghe) Company Limited is selected for the test. Ultra-fine cement (UFC) is prepared by the continuous and combined action of OPC P·O42.5 through the constant and combined action of mechanical forces such as impact, grinding, extrusion, and bending.

**Table 1.** Physical properties of soil.

| Test Soil | Natural Moisture Content (%) | Plastic Limit (%) | Liquid Limit (%) | Natural Density (g·cm$^{-3}$) | Specific Gravity of Soil Particle (Gs) |
|---|---|---|---|---|---|
| Cohesive soil | 18.6 | 23.0 | 39.2 | 1.96 | 2.73 |

Tables 2 and 3 are the physical index and main chemical component content data of the cement used in the test. Data analysis shows that UFC has a smaller particle size and specific surface area than OPC, and the main chemical components and contents of the two are basically the same.

**Table 2.** Physical index of cement binders.

| | $D_{50}$/µm | $D_{95}$/µm | Maximum Grain/µm | Specific Surface Area/(cm$^2$/g) |
|---|---|---|---|---|
| OPC | 18.05 | 77.16 | 138.04 | 7880 |
| UFC | 3.13 | 8.23 | 11.83 | 10,703 |

**Table 3.** Main chemical components of cement binders.

| Material Type | Chemical Composition and Its Mass Fraction (%) | | | |
|---|---|---|---|---|
| | CaO | SiO$_2$ | Al$_2$O$_3$ | Fe$_2$O$_3$ |
| OPC | 65.5 | 18.4 | 5.3 | 2.9 |
| UFC | 65.5 | 18.0 | 5.4 | 2.9 |

Figure 1a,b,e,f are SEM images of cohesive soil aggregates, HA aggregates, OPC, and UFC. The aggregate structure of cohesive soil is relatively compact and mainly connected by surface-surface contact. HA agglomerates are formed by accumulating particles of different sizes, and the pores are interconnected and have a sponge-like structure. Compared with OPC particles, UFC particles have better roundness and smaller particles.

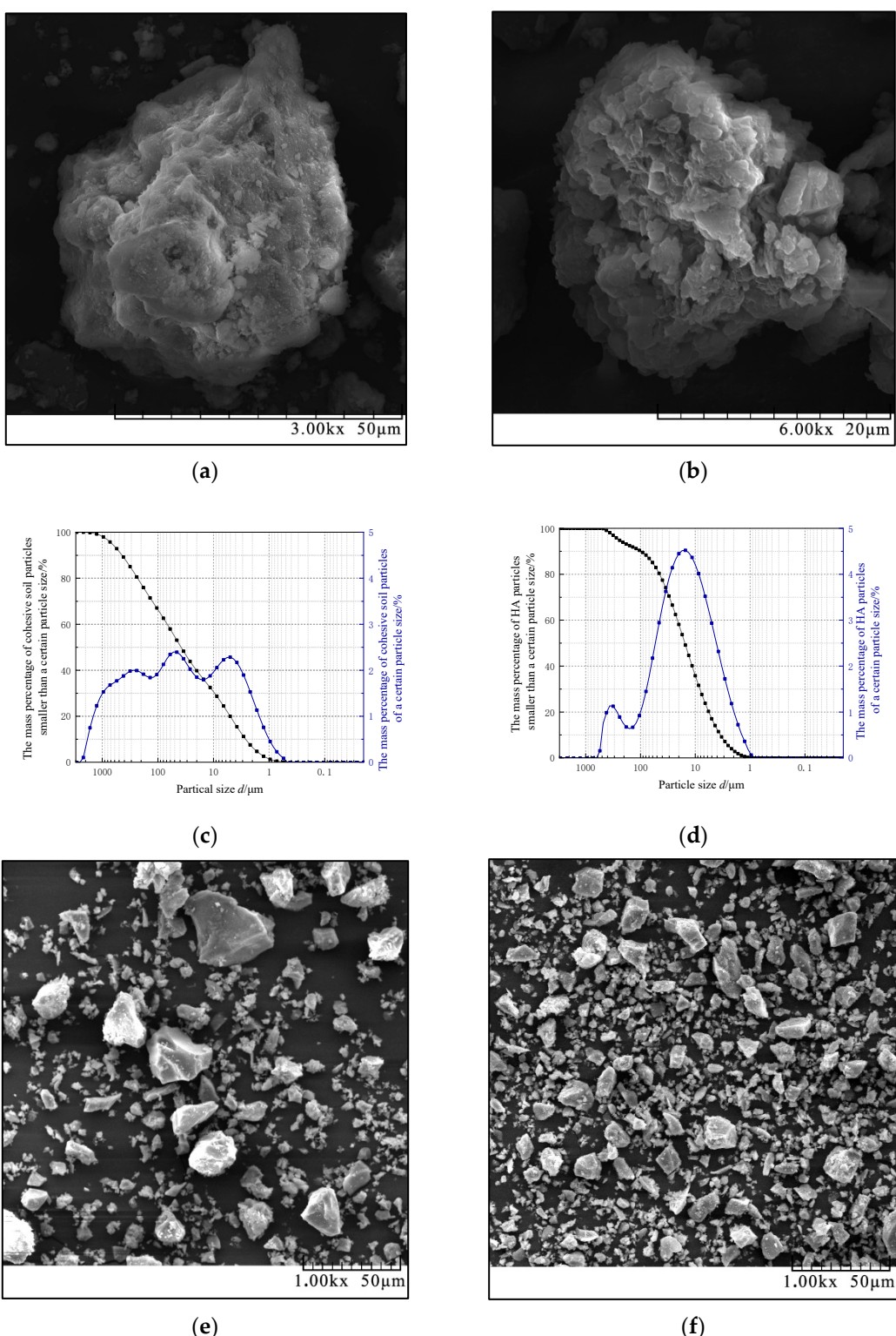

**Figure 1.** *Cont.*

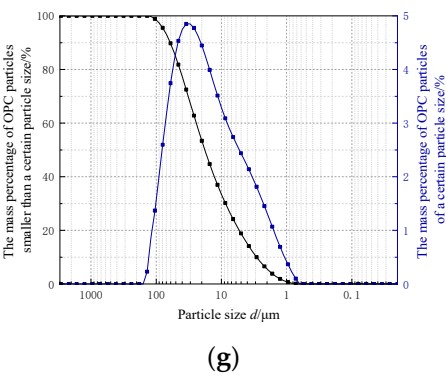
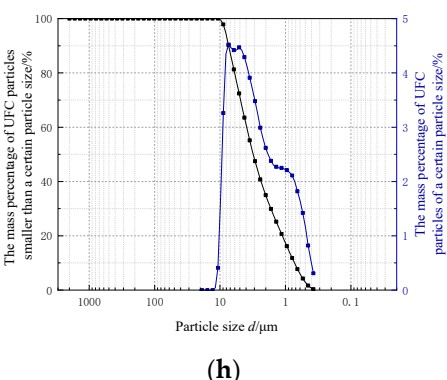

(**g**)                                      (**h**)

**Figure 1.** Characteristic of test material. (**a**)SEM image of cohesive soil aggregates. (**b**) SEM images of HA aggregates. (**c**) PSD of cohesive soil. (**d**) PSD of HA. (**e**) SEM image of OPC. (**f**) SEM image of UFC. (**g**) PSD of OPC. (**h**) PSD of UFC.

Figure 1c,d,g,h correspond to the particle size distribution (PSD) of cohesive soil, HA, OPC, and UFC, respectively. The particle size of the soil used in the test is less than 0.075 mm, the content of soil particles exceeds 60% of the total weight, and the plasticity index $I_p > 10$. The soil used for the test is classified as cohesive soil according to the engineering classification of "Code for design of building foundation" (GB 50007-2011). The specific surface area of OPC is greater than 3000 cm$^2$/g, which meets the cement fineness requirements in "Common Portland cement" (GB 175-2007). The maximum particle size of UFC is less than 12 µm, $D_{95} < 10$ µm, the median particle size is less than 5 µm, and the specific surface area is greater than 8000 cm$^2$/g, which is in line with the definition of UFC in domestic and foreign research.

### 2.2. Experimental Method

The test set the water-cement ratio ($c = 0.5$), water content by mass ($w = 24\%$), void ratio ($e = 0.8$), HA content ($\lambda = 15\%$), and cement rate ($\beta = 20\%$) of the sample. The proportion of UFC is 0%, 10%, 20%, 30%, 40%, and 50%, and the soaking time is 90 d. The samples are subjected to mercury intrusion porosimetry (MIP), scanning electron microscopy (SEM), X-ray powder diffraction (XRD), and an unconfined compressive strength test (UCS).

The samples are made according to the "Standard for geotechnical testing method" (GB/T 50123-2019). The amounts of cohesive soil particles, HA particles, OPC, UFC, and distilled water are calculated according to formulas (1) to (4). After the test materials are thoroughly mixed, 1/3 of the mass of a single sample is poured into three valves with an inner diameter of d = 39.1 mm and a height of h = 80.0 mm for compaction. The compaction height of each layer is 26.7 mm, and the surface at the junction of each layer is roughening. After demolding, the samples are sealed with plastic wrap, placed in a curing box, and maintained at a temperature of 20 ± 3 °C for 10 days. The samples were soaked in distilled water after curing.

$$\beta = \frac{m_{s(OPC)} + m_{s(UFC)}}{m_{s(HA)} + m_{s(soil)}} \times 100\% \tag{1}$$

$$\gamma = \frac{m_{s(UFC)}}{m_{s(OPC)} + m_{s(UFC)}} \times 100\% \tag{2}$$

$$\lambda = \frac{m_{s(HA)}}{m_{s(HA)} + m_{s(soil)}} \times 100\% \tag{3}$$

$$m_{(W)} = \left( m_{s(HA)} + m_{s(soil)} \right) \times \omega + \left( m_{s(OPC)} + m_{s(UFC)} \right) \times c \tag{4}$$

where $m_{s(OPC)}$ is the OPC particle weight, $m_{s(UFC)}$ is the UFC particle weight, $m_{s(HA)}$ is the HA particle weight, $m_{s(soil)}$ is the cohesive soil particle weight, $m_{(W)}$ is the distilled water weight, $\omega$ is the water content by mass, and c is the water-cement ratio.

### 2.3. Experimental Devices

In this paper, MIP, SEM, XRD, and UCS tests are performed on the samples, respectively, and the test steps are shown in Figure 2. Before performing MIP, SEM, and XRD, take a parallel sample and bake it to a constant weight in an oven at a drying temperature of 50 °C. The sample is divided into three equal parts along the height, and the bulk and powder samples are made, respectively. Take two-thirds of the sample to make a block sample, use tweezers to pick up a suitable block as a block sample, and do not directly touch the sample during sample preparation. Then, 1~2 g bulk samples are taken, and MIP is performed using the Autopore9510 high-performance automatic mercury porosimeter produced by Micromeritice in the United States [38–40]. Take a block sample with a length and width of about 5~8 mm and fix it on the sample stage with conductive glue, blow off the dust on the sample surface, and put it into the electron microscope after being sprayed with gold. The SEM test s carried out with the Czech TESCAN-VEGA3 electron microscope [39–41]. One-third of the sample is taken to make a powder sample. The bulk sample is crushed with a pestle, ground, and sieved clockwise to prepare a powder with a particle size of about 5 μm [39]. Then, 1~2 g powder samples are taken into Holland PANalytical X'Pert3 Powder type multifunctional powder X-ray diffractometer for XRD. Take samples from distilled water for UCS. The samples are placed in a cool place and air-dried to the quality at the time of sample preparation, and UCS is carried out using the CSS-44000 electronic universal testing machine of Changchun Testing Machine Research Institute. The axial compression rate of the testing machine is 1.0 mm/min.

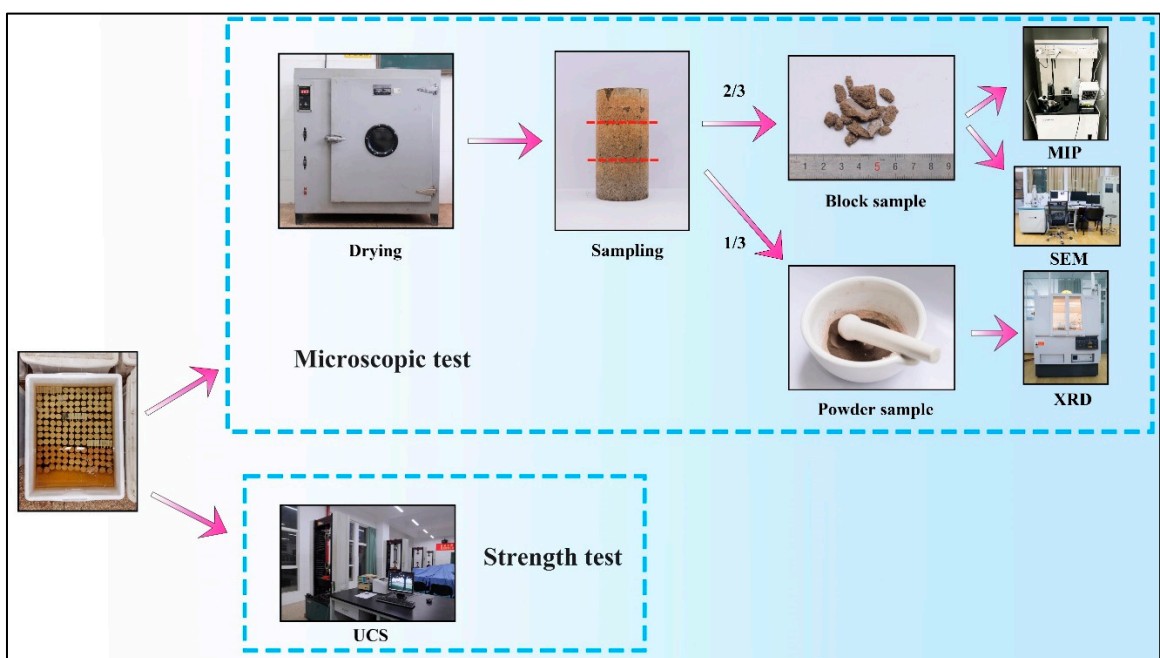

**Figure 2.** Experimental process.

## 3. Microscopic Test Results and Analysis

### 3.1. MIP Results and Analysis

Figure 3 is the pore size distribution curve of cemented soil containing humic acid (HA) under different proportions of ultra-fine cement (UFC). The pore size of the pores is divided into 4 categories according to the curve characteristics of Figure 3a,b: macropores, pore size d > 10 μm; mesopores, pore size 0.1 μm < d < 10 μm; small pores, pore size 0.01 μm < d < 0.1 μm; micropores, pore size d < 0.01 μm. Figure 3a shows the percentage distribution of pore volume with different pore sizes in the total pore volume of HA-containing cemented soil under different proportions of UFC. The value of the peak point of the peak-shaped distribution curve gradually decreases with the increase in the proportion

of UFC, and the distribution area gradually decreases. The value of the peak point of the peak-shaped distribution curve changed significantly when the proportion of UFC increased from 0% to 30%. Figure 3b shows that with the increase in UFC proportion, the cumulative volume of pores of the sample decreases continuously. The sample without UFC has the largest cumulative pore volume, and the sample with UFC proportion of 50% has the smallest cumulative pore volume.

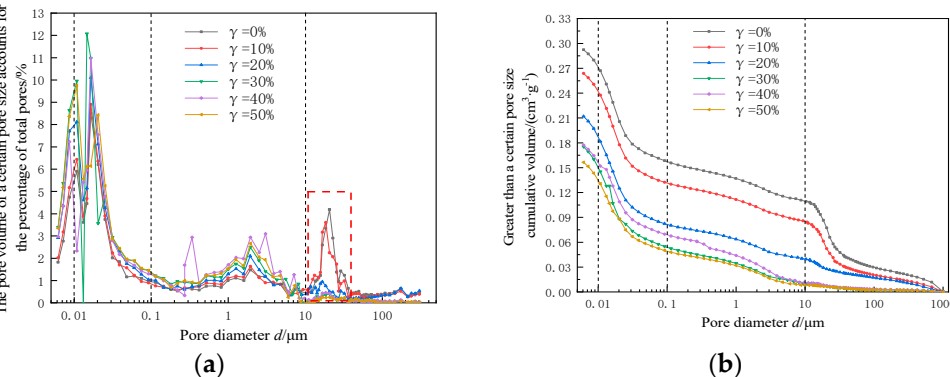

**Figure 3.** Pore size distribution curve of HA-containing cemented soil under different proportions of UFC. (**a**) Pore volume percentage distribution curve. (**b**) Pore volume accumulation curve larger than a certain pore size.

Ordinary Portland cement (OPC) is composed of tricalcium silicate ($C_3S$), dicalcium silicate ($C_2S$), tricalcium aluminate ($C_3A$), and tetra calcium ferric aluminate ($C_4AF$). $C_3A$, $C_3S$, and $C_4AF$ undergo rapid hydration reactions upon contact with water. The hydration reactions of the four main cement clinker minerals are represented by Equations (5)–(8). The hydration reaction produces calcium silicate hydrate (C-S-H) gels with a certain strength, calcium aluminate hydrate, calcium sulfoaluminate hydrates, and calcium hydroxide crystals. Generally, a small amount of gypsum is added to OPC to avoid some negative effects caused by the fast hydration reaction of $C_3A$. Gypsum's presence causes the formula's chemical reaction (9) to form ettringite (AFt). The hydration product cements the soil particles and fills the pores between the particles. HA particles are partially dissolved in the alkaline environment formed by the hydration reaction, which affects the formation of hydration products [42]. The increase in the proportion of UFC makes the macropores filled with hydration products first, so the percentage of macropore volume in the total pores gradually decreases. According to Power's theory [43–46], the increase in hydration products reduces the total pore volume of the sample. The increase in the proportion of UFC increases the content of small particles and the specific surface area of the cement, the contact area between the cement particles and the solution increases, and the hydration reaction speed is accelerated. The increase in the proportion of UFC increases the content of small particles and the specific surface area of the cement, the contact area between the cement particles and the solution increases, and the hydration reaction speed is accelerated.

$$2(3CaO \cdot SiO_2) + 6H_2O \rightarrow 3CaO \cdot 2SiO_2 \cdot 3H_2O + 3Ca(OH)_2 \tag{5}$$

$$2(2CaO \cdot SiO_2) + 4H_2O \rightarrow 3CaO \cdot 2SiO_2 \cdot 3H_2O + Ca(OH)_2 \tag{6}$$

$$3CaO \cdot Al_2O_3 + 6H_2O \rightarrow 3CaO \cdot Al_2O_3 \cdot 6H_2O \tag{7}$$

$$4CaO \cdot Al_2O_3 \cdot Fe_2O_3 + 7H_2O \rightarrow 3CaO \cdot Al_2O_3 \cdot 6H_2O + CaO \cdot Fe_2O_3 \cdot H_2O \tag{8}$$

$$3CaO \cdot Al_2O_3 \cdot 6H_2O + 3(CaSO_4 \cdot 2H_2O) + 19H_2O \rightarrow 3CaO \cdot Al_2O_3 \cdot 3CaSO_4 \cdot 31H_2O \tag{9}$$

Figure 4 shows the volume percentage of macropores (>10 μm) of HA-containing cemented soil under different UFC proportions. The volume percentage of macropores with UFC proportion of 30% is reduced by about 28.12% compared to the sample without UFC.

HA particles and cohesive soil particles together constitute the skeleton of cemented soil. Since the particle size of HA is between 10 μm and 100 μm, which accounts for nearly 50%, it will be dissolved in a small amount in the alkaline environment generated by the hydration reaction so that the HA-containing cemented soil forms a macropore structure [42]. The interaction of HA with the hydration reaction leads to the reduction of hydration products. When the content of HA and the cement rate dosing remain unchanged, the increase in the proportion of UFC speeds up the hydration reaction rate, and the increase in hydration products gradually reduces the macropores in the sample. Therefore, incorporating UFC into cement can inhibit the effect of HA on cemented soil.

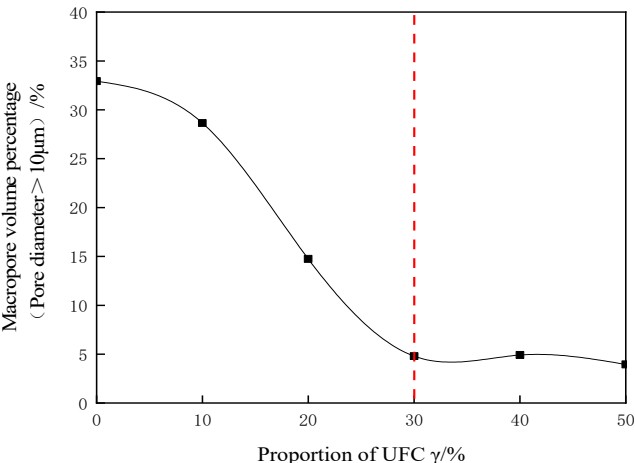

**Figure 4.** Percentage of macropore volume in total pores of HA-containing cemented soil under different UFC proportions.

Figure 5 shows the porosity of HA-containing cemented soil under different UFC proportions. Compared with the HA-containing cemented soil without UFC, it is found that the incorporation of UFC significantly reduces the porosity of the sample. The porosity of the samples decreased significantly when the proportion of UFC increased from 0% to 30%. The porosity with UFC proportion of 30% is reduced by about 40.01% compared to the sample without UFC. The trends of the two curves in Figures 4 and 5 are basically the same, reflecting that macropores are an important part of the pore structure of HA-containing cemented soil. The volume percentage of macropores and the porosity are correlated well. The volume percentage of macropores affects the structure of HA-containing cemented soil.

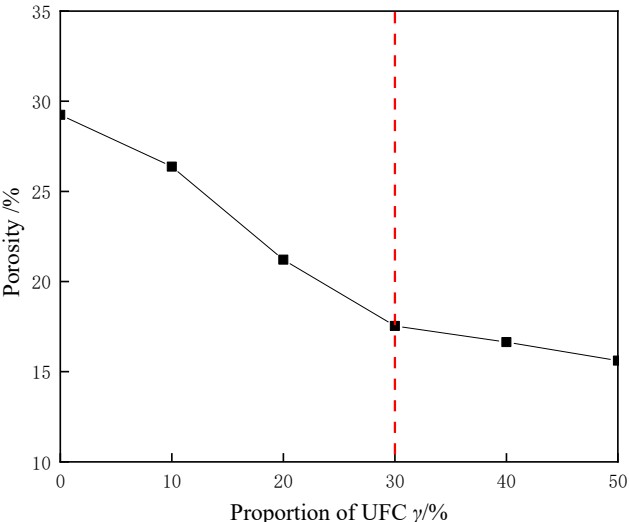

**Figure 5.** Porosity of HA-containing cemented soil under different proportions of UFC.

## 3.2. SEM Results and Analysis

Figure 6 is the SEM images of 500 and 2000 times of HA-containing cemented soil under different UFC proportions. Figure 6a is an SEM image of HA-containing cemented soil without UFC, which has the following characteristics: (1) There are obvious structural elements in the sample. The contact state and connection form of the structural elements are loose, the pores between the units are connected, and the structure is loose and overhead. (2) Fibrous calcium silicate hydrate and acicular ettringite are scattered in the pores, forming a unique space grid structure to support and fill the pores.

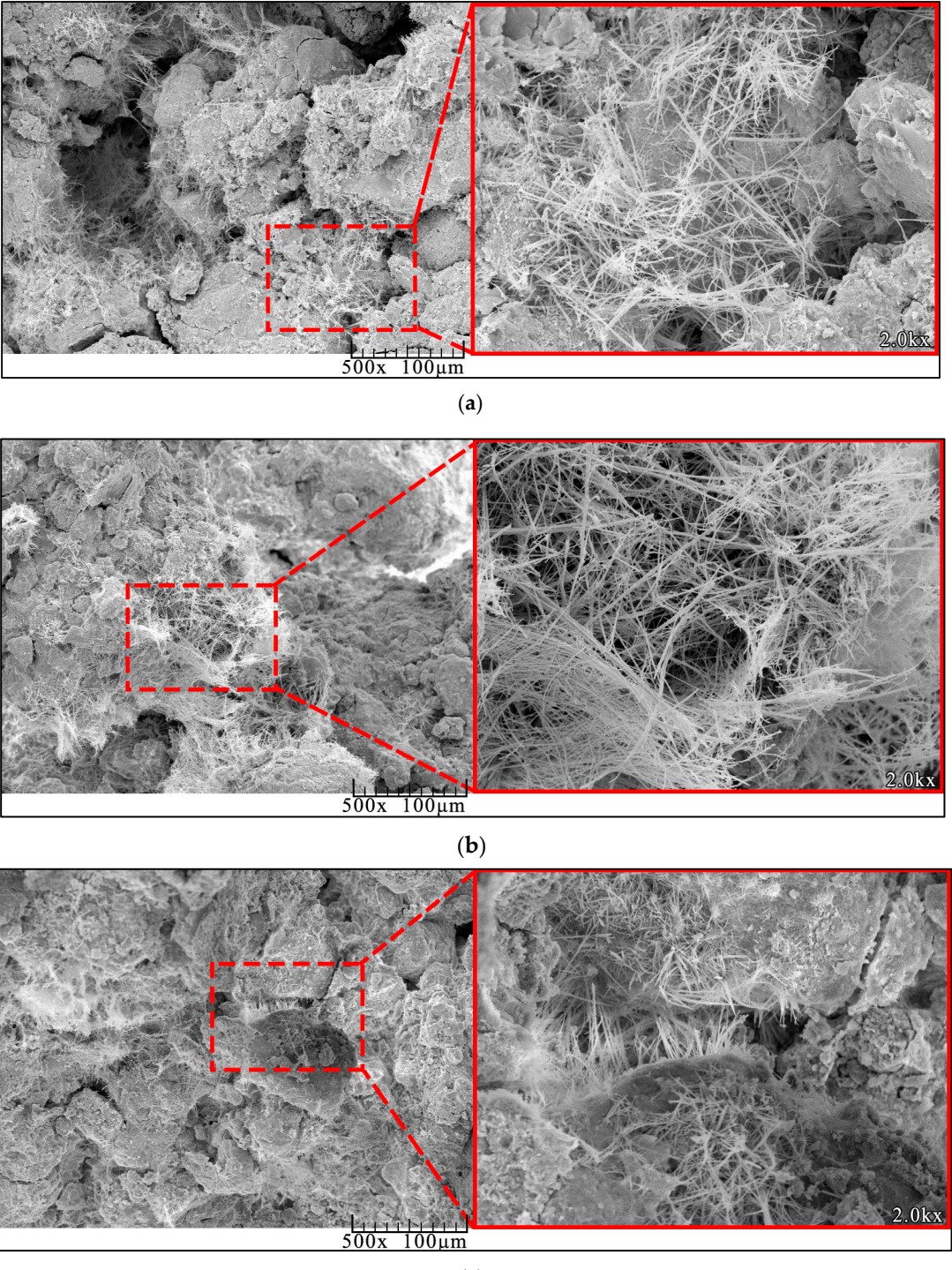

**Figure 6.** *Cont.*

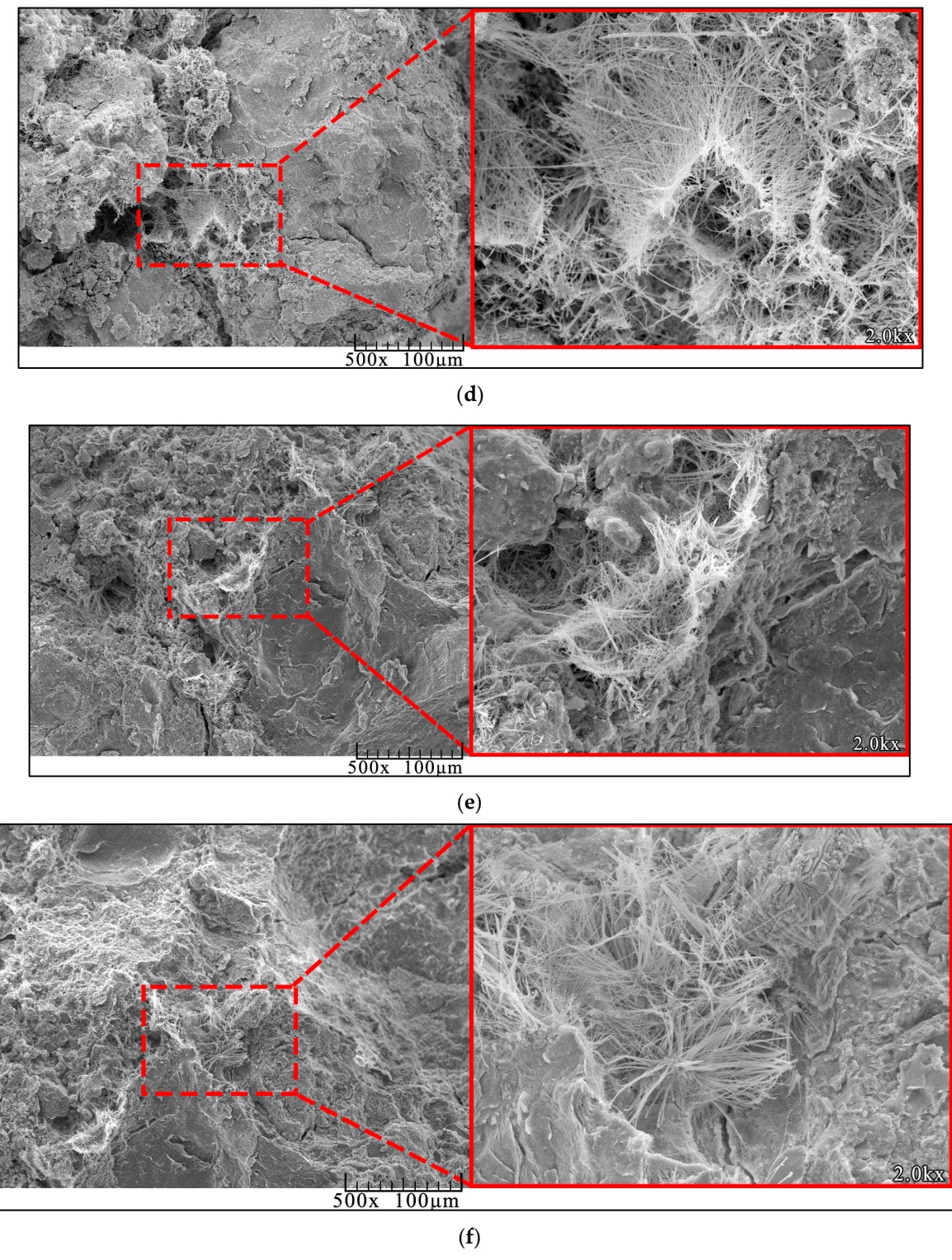

**Figure 6.** SEM images of HA-containing cemented soil samples under different proportions of UFC. (**a**) $\gamma = 0\%$. (**b**) $\gamma = 10\%$. (**c**) $\gamma = 20\%$. (**d**) $\gamma = 30\%$. (**e**) $\gamma = 40\%$. (**f**) $\gamma = 50\%$.

Figure 6b–f are SEM images of HA-containing cemented soil with UFC proportion of 10%~50%, which have the following characteristics: (1) There is no obvious through-hole on the surface of the sample. (2) The number of macropores gradually decreases with the increase in the proportion of UFC. The pores are changed from fibrous filling to cemented filling, the volume of the structural elements composed of hydration products and aggregates increases, and the structure of HA-containing cemented soil improves.

(3) The cementation effect of the hydration product is obvious when the proportion of UFC is greater than 30%, which makes the connection between the structural elements closer and the structural integrity better.

UFC gradually replaces ordinary Portland cement (OPC) when the cement rate is unchanged. The increased specific surface area and chemical activity of the cement particles make the hydration reaction faster. The enhanced filling and cementation of hydration products improved the microscopic pore structure and the HA-containing cemented soil structure. Comparing the SEM images of undoped UFC and HA-containing cemented soil samples with a UFC proportion of 10%, it is found that the incorporation of UFC resulted in no obvious through pores between the units. The soil sample prepared from HA and cohesive soil is a multiphase dispersion system that exhibits general colloidal characteristics when combined with water. The most abundant silica in the soil sample forms colloidal silicic acid particles in water, with $Na^+$ or $K^+$ on the surface. These ions are adsorbed and exchanged with the equivalent of $Ca^{2+}$ generated by cement hydration so that smaller soil particles form aggregates. The composite cement curing agent mixed with UFC has higher chemical activity than OPC, and the gel particles generated by its hydration reaction have a larger specific surface area, thus generating larger surface energy and strong adsorption activity. The gel particles combine larger aggregates to form the structural element of cemented soil and close the pores between the structural element. The hydration reaction speed accelerated with the increase in the proportion of UFC, which further accelerated the connection of the aggregates and the structural element volume further increased. The boundary between the units disappears when the proportion of UFC is 50%, there are discontinuous and dispersed small pores on the surface of the sample, and the structure is dense.

Comparing the results of the mercury intrusion test (MIP), it can be seen that the increase in the proportion of UFC increases the hydration degree of the cement. The enhanced cementation of the hydration products makes the pores change from fibrous filling to cemented filling, and the number and diameter of the pores continue to decrease. The larger aggregates were further combined to form structural elements, which enhanced the integrity of the HA-containing cemented soil. The porosity of the sample decreases slowly when the proportion of UFC is greater than 30%. It can be seen from Figure 6e,f that the pores of the sample are basically filled with fibrous hydration products, and the hydration reaction is limited by space, so the porosity of the sample decreases slowly.

*3.3. XRD Results and Analysis*

In this paper, Jade 6.0 software is used as the analysis software for X-ray powder diffraction (XRD) results, and the peak intensity of the test curve is compared with the peak intensity of the card. Figure 7 is the XRD pattern of cohesive soil, and its main components are: (1) original mineral, quartz ($SiO_2$) and mica ($KAl_2(AlSi_3O_{10})(OH)_2$); (2) secondary mineral, kaolinite ($Al_2O_3 \cdot 2SiO_2 \cdot 2H_2O$). Figure 8 shows the XRD patterns of HA-containing cemented soil corresponding to UFC proportion $\gamma$ of 0%, 10%, 20%, 30%, 40%, and 50%, and its main components are (1) original mineral, quartz ($SiO_2$) and mica ($KAl2(AlSi_3O_{10})(OH)_2$); (2) secondary mineral, kaolinite ($Al_2O_3 \cdot 2SiO_2 \cdot 2H_2O$); (3) hydration products, calcium silicate hydrate gel (C-S-H) and tobermorite ($Ca_5Si_6O_{16}(OH)_2 \cdot 8H_2O$).

The phase analysis of the XRD patterns of the HA-containing cemented soil samples under different UFC proportions in Figure 8 shows that no new characteristic peaks appear except for the known phases. The crystal structure of humic acid is controversial, so no further analysis is performed [47]. According to the relevant research results of scholars [48–51], the hydration products of cement are mainly calcium silicate hydrate gel (C-S-H, $2\theta$ are 29.1°, 31.8°, 49.8°, and 54.9°, etc.) and tobermorite ($Ca_5Si_6O_{16}(OH)_2 \cdot 8H_2O$, $2\theta$ are 16.1°, 29°, and 49.6°, etc.). The characteristic peak heights of hydrated calcium silicate gel and tobermorite increased slowly with UFC proportion, and the relative content of hydration products increased slightly. Relevant studies have shown that the complete hydration of 1 $cm^3$ cement requires a space of 2 $cm^3$ to accommodate the hydration prod-

ucts [44]. SEM results show that the number of pores in the sample decreases when the proportion of UFC increases from 0% to 50%, the space provided for hydration products gradually decreases, and the formation of hydration products is affected.

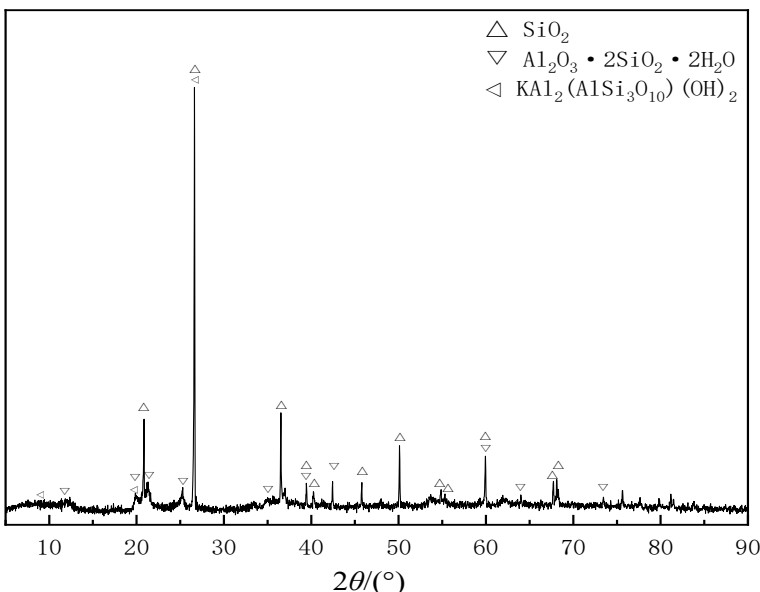

**Figure 7.** XRD pattern of cohesive soil.

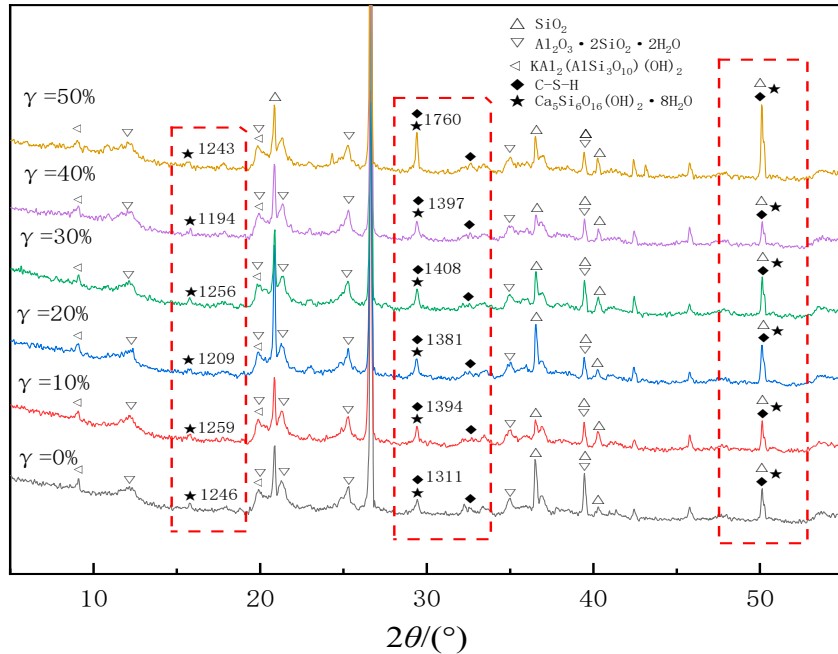

**Figure 8.** XRD patterns of HA-containing cemented soil samples under different UFC proportions.

## 4. Strength Test Results and Analysis

It can be seen from Figure 9 that the unconfined compressive strength $q_u$ of cemented soil mixed with humic acid (HA) increases gradually with the increase in the proportion of ultra-fine cement (UFC). The strength of cemented soil is mainly affected by two aspects: (1) cementation of hydration products and (2) the cemented soil's porosity, pore size, and connectivity [52,53]. Mercury intrusion porosimetry (MIP), scanning electron microscopy (SEM), and X-ray powder diffraction (XRD) results showed that the chemical activity of composite cement curing agent increased with the proportion of UFC. The increase in

chemical activity makes the hydration reaction faster and produces more hydration products. The filling and cementation of hydration products reduce the pore volume in the HA-containing cemented soil, improve the loose and overhead structure, and gradually reduce the volume of macropores. Combined with the analysis of Figure 6a,b of the SEM test, the enhanced cementation of the hydration product significantly reduces the pores in the sample, improving the strength. Consequently, the filling and cementation effect of the hydration products is enhanced with the increase in the UFC proportion. The total pore volume of the sample gradually decreased, the volume percentage of macropores and porosity gradually decreased, and the connectivity of the pores weakened. The improvement of microscopic pore structure makes the strength of HA-containing cemented soil show a gradually increasing trend. Under the same condition of unconfined compressive strength $q_u$, the cement consumption decreased by 10.4%~20.1% when the proportion of UFC increased from 10% to 50%.

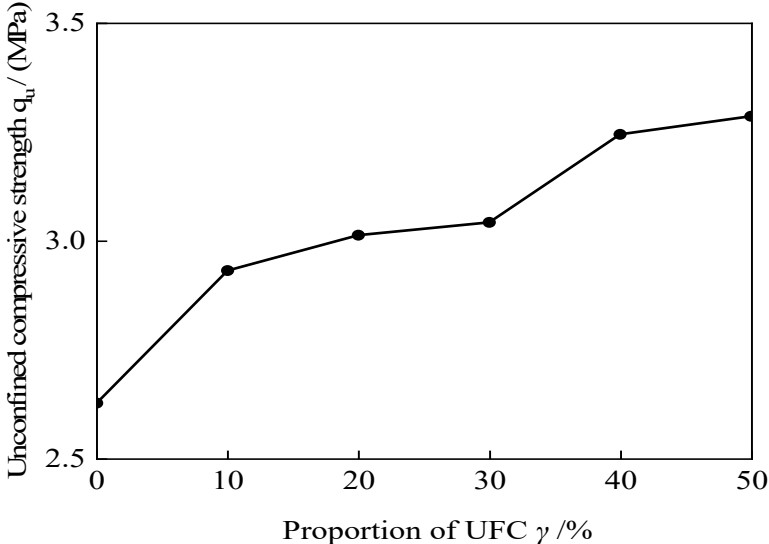

**Figure 9.** Unconfined compressive strength curves of HA-containing cemented soil with different UFC proportions.

## 5. Conclusions

In this study, microscopic and strength tests are carried out on cemented soil containing humic acid (HA) with different proportions of ultra-fine cement (UFC). Based on the microscopic test and strength test results, this paper explores the effect mechanism of UFC on the microscopic pore structure of HA-containing cemented soil. The following conclusions are drawn:

1.  The mercury intrusion porosimetry (MIP) results showed that the hydration reaction speed increased with the increase in the proportion of UFC, and the increase in hydration products could fill the pores of HA-containing cemented soil. The volume percentage of macropores decreased significantly when the proportion of UFC increased from 0% to 30%. The percentage of macropores in the sample correlates well with the porosity, and the reduction of macropores enhances the HA-containing cemented soil structure.

2.  Scanning electron microscopy (SEM) showed that the increase in the proportion of UFC would accelerate the hydration rate, and the HA-containing cemented soil structure would tend to be compact from loose and overhead. The aggregates are continuously combined to form a structural element, and the increase in the proportion of UFC makes the connectivity of the structural element stronger.

3.  X-ray powder diffraction (XRD) showed that although the increase in the proportion of UFC accelerated the hydration reaction and generated more hydration products, the formation of hydration products is affected due to the limitation of pore space.

The relative content of hydration products increased slightly with the increase in the proportion of UFC.

4.  UCS shows that the strength of HA-containing cemented soil increases gradually with UFC proportion. The strength of HA-containing cemented soil increased significantly when the proportion of UFC increased from 0% to 10%. This is because the addition of UFC enhances the cementation of cement filling and reduces the permeability of pores.

5.  The analysis of the microscopic and strength test results shows that incorporating UFC can reduce HA's influence on cemented soil. In the related research and practical engineering of peat soil in the Dianchi Lake area, the conclusions of this paper can be used for in-depth research.

6.  Through further research, the purpose of reducing the total amount of cement when solidifying the peat soil foundation can be achieved, providing a theoretical basis for reducing $CO_2$ emissions in the project and promoting the sustainable development of society.

**Author Contributions:** Conceptualization, J.C. and F.L.; methodology, J.C.; formal analysis, F.L. and S.H.; resources, Z.S., J.L. and G.L.; data curation, F.L.; writing—original draft preparation, F.L.; writing—review and editing, H.L.; visualization, J.C.; project administration, Z.S.; funding acquisition, J.C. All authors have read and agreed to the published version of the manuscript.

**Funding:** This research was funded by Natural Science Foundation of Yunnan Province (China), grant number 41967035.

**Institutional Review Board Statement:** Not applicable.

**Informed Consent Statement:** Not applicable.

**Data Availability Statement:** The data used to support the finding of this study are included in the article.

**Conflicts of Interest:** The authors declare no conflict of interest.

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
