# Peer review of "Effect of UFC on the Microscopic Pore Structure of Cemented Soil in Humic Acid Environment"

_sustainability, doi:10.3390/su15043241_

Round 1

Reviewer 1 Report

Dear authors!

The subject of the manuscript is relevant both for the region of study and for other regions. The manuscript presents quite interesting results of the original study. However, there are a number of important comments. I recommend the authors to correct the article.

1. The data about reducing CO2 emissions should be considered in more detail.

2. P.2 l.91-94 the phrase "Therefore, the cement-soil mixing method is one of the technical means to strengthen the peat soil foundation" is given twice.

3. The text lacks a clearly defined goal and objectives of the study. I recommend citing them at the end of the Introduction (P.2 l.97).

4. In Materials and Methods (P.3 l.107, l.109) in the description of cement is "OPC 42.5#". The authors apparently meant OPC 42.5 R or OPC 42.5 n.

5. P.5 l.152 Table 4 is not needed. It duplicates the text given above (P.5 l.146-151).

6. P.6 l.165-166. Figure 2 is redundant. It does not carry significant information for the study.

7. P.7 l.206-214, P.8 equations 5-9 the authors discuss the processes of hydration of OPC components with water. However, they do not indicate the presence of HA, which change the acidity of the aqueous phase and can react with OPC components. Below (P.8 l.233-240) the authors discuss the influence of hydration products on HA, while the influence of HA on hydration processes is not considered. This aspect should also be taken into account.

8. P.8 l.239-241 The phrase "Therefore, incorporating UFC into cement can inhibit the effect of HA on cement-soil" appears twice.

9. P.9 l.253-256 The phrase “The volume percentage of macropores affects the structure of HA-containing cement-soil. ” is given twice and duplicates the meaning of the previous sentence.

10. In Conclusions (P.13 l.395-399) in conclusion 5, the first two sentences are redundant. Their meaning is duplicated below.

11. In Conclusions (P.13-14 l.402-406) I propose to move the last sentence in Conclusion 5 to a separate paragraph.

Author Response

Response to Reviewer 1 Comments

Many thanks to the reviewers for their suggestions on our manuscript. The revision comments were very professional and considerably improved our manuscript. We have carefully reviewed the comments and have revised the manuscript accordingly. In particular, We re-analyze the UFC's role in reducing CO2 emissions. Below, I will reply to the revision comments one by one.

Point 1: The data about reducing CO2 emissions should be considered in more detail.

Response 1: We analyzed the data and added new analyses. Under the condition that the cement-soil strength is consistent, the addition of UFC can reduce the amount of cement.(Make revisions in P.12 l.363-365 of the manuscript.)

Point 2: P.2 l.91-94 the phrase "Therefore, the cement-soil mixing method is one of the technical means to strengthen the peat soil foundation" is given twice.

Response 2: We have removed the redundant sentence " Therefore, the cement-soil mixing method is one of the technical means to strengthen the peat soil foundation ".(Make revisions in P.2 l.92-95 of the manuscript.)

Point 3: The text lacks a clearly defined goal and objectives of the study. I recommend citing them at the end of the Introduction (P.2 l.97).

Response 3: We supplement the research aims and objectives in the last paragraph of the Introduction.(Make revisions in P.3 l.97-99 of the manuscript.)

Point 4: In Materials and Methods (P.3 l.107, l.109) in the description of cement is "OPC 42.5#". The authors apparently meant OPC 42.5 R or OPC 42.5 n.

Response 4: Thank you for your reminder. The type of cement used in the test should be "Ordinary Portland Cement P·O 42.5". (Make revisions in (P.3 l.109, l. 111) of the manuscript.)

Point 5: P.5 l.152 Table 4 is not needed. It duplicates the text given above (P.5 l.146-151).

Response 5: We removed " Table 4. Test scheme ". (Make revisions in P.5 l.153-154 of the manuscript.)

Point 6: P.6 l.165-166. Figure 2 is redundant. It does not carry significant information for the study.

Response 6: We removed " Figure 2. Sample production process ". (Make revisions in P.5 l.165-166 of the manuscript.)

Point 7: P.7 l.206-214, P.8 equations 5-9 the authors discuss the processes of hydration of OPC components with water. However, they do not indicate the presence of HA, which change the acidity of the aqueous phase and can react with OPC components. Below (P.8 l.233-240) the authors discuss the influence of hydration products on HA, while the influence of HA on hydration processes is not considered. This aspect should also be taken into account.

Response 7: HA particles will be partially dissolved in the alkaline environment formed by the hydra-tion reaction, which will affect the formation of hydration products. (Make revisions in P.7 l.214-216 of the manuscript.)

The interaction of HA with the hydration reaction leads to the reduction of hydration products. (Make revisions in P.7 l.238-239 of the manuscript.)

Point 8: P.8 l.239-241 The phrase "Therefore, incorporating UFC into cement can inhibit the effect of HA on cement-soil" appears twice.

Response 8: We have removed the redundant sentence "Therefore, incorporating UFC into cement can inhibit the effect of HA on cement-soil".(Make revisions in P.7 l.241-242 of the manuscript.)

Point 9: P.9 l.253-256 The phrase “The volume percentage of macropores affects the structure of HA-containing cement-soil.” is given twice and duplicates the meaning of the previous sentence.

Response 9: We have removed the redundant sentence " The volume percentage of macropores affects the structure of HA-containing cement-soil". (Make revisions in P.8 l.255-256 of the manuscript.)

Point 10: In Conclusions (P.13 l.395-399) in conclusion 5, the first two sentences are redundant. Their meaning is duplicated below.

Response 10: We have removed the first two sentences in Conclusion 5. (Make revisions in P.13 l.397-400 of the manuscript.)

Point 11: In Conclusions (P.13-14 l.402-406) I propose to move the last sentence in Conclusion 5 to a separate paragraph.

Response 11: We change the last sentence of Conclusion 5 into Conclusion 6. (Make revisions in P.13 l.401-404 of the manuscript.)

Reviewer 2 Report

The authors could give more information about Peat soil and its physical, chemical, and other properties.

Author Response

Response to Reviewer 2 Comments

We sincerely thank the editor and all reviewers for their valuable feedback that we have used to improve the quality of our manuscript. We have carefully reviewed the comments and have revised the manuscript accordingly. The following are the corresponding revisions.

Point 1: The authors could give more information about Peat soil and its physical, chemical, and other properties.

Response 1: Peat soil has a large natural void ratio, high natural water content, high organic matter content, large compressibility coefficient, low shear strength, and low bearing capacity. (Make revisions in P.1 l.37-39 of the manuscript.)

Reviewer 3 Report

line 146-147 : The test controls the water-cement ratio c=0.5, the water content w=24%, the void ratio e=0.8, the designed HA content 15%, and the cement rate 20%. - to be rephrase/revise

Author Response

Response to Reviewer 3 Comments

We have carefully studied the valuable comments from the editor and all reviewers and tried our best to revise the manuscript. Many thanks to the reviewers for their suggestions on our manuscript. The point-to-point responses to the reviewer's comments are listed as follows.

Point 1: line 146-147 : The test controls the water-cement ratio c=0.5, the water content w=24%, the void ratio e=0.8, the designed HA content 15%, and the cement rate 20%. - to be rephrase/revise.

Response 1: The test set the water-cement ratio (c=0.5), water content by mass (w = 24%), void ratio (e = 0.8), HA content (λ=15%), and cement rate (β=20%) of the sample. (Make revisions in P.5 l.149-150 of the manuscript.)
